# In Vivo Effect of Resveratrol-Loaded Solid Lipid Nanoparticles to Relieve Physical Fatigue for Sports Nutrition Supplements

**DOI:** 10.3390/molecules25225302

**Published:** 2020-11-13

**Authors:** Lili Qin, Tianfeng Lu, Yao Qin, Yiwei He, Ningxin Cui, Ai Du, Jingyu Sun

**Affiliations:** 1Sports and Health Research Center, Department of Physical Education, Tongji University, Shanghai 200092, China; qinlili@tongji.edu.cn (L.Q.); 02110@tongji.edu.cn (T.L.); 1931799@tongji.edu.cn (Y.Q.); 1831832@tongji.edu.cn (Y.H.); 2031825@tongji.edu.cn (N.C.); 2School of Physics Science and Engineering, Tongji University, Shanghai 200092, China; duai@tongji.edu.cn

**Keywords:** solid lipid nanoparticles (SLNs), resveratrol, drug delivery, anti-fatigue, nutrition supplements

## Abstract

Resveratrol (RSV) is a natural flavonoid polyphenol compound extracted from the plants which shows various biological activities. However, the clinical application of RSV is limited by its poor aqueous solubility, rapid metabolism and poor bioavailability. In this study, resveratrol-loaded solid lipid nanoparticles (RSV- SLNs) was design as a nano-antioxidant against the physical fatigue. The resultant RSV-SLNs were characterized by photon correlation spectroscopy (PCS), transmission electron micrographs (TEM), zeta potential, differential scanning calorimetry (DSC) and Raman spectroscopy pattern. Furthermore, the in vivo anti-fatigue effect assays showed that RSV-SLNs prolonged the mice exhausted time and running distance. The biochemical parameters of blood related to fatigue suggested that RSV-SLNs have potential applications to improve the antioxidant defense of the mice after extensive exercise and confer anti-fatigue capability. Furthermore, the molecular mechanisms of antioxidant by RSV-SLNs supplementation was investigated through the analysis of silent information regulator 2 homolog 1 (SIRT1) protein expression, which demonstrated that it could downregulate the expression of SIRT1 and increase autophagy markers, microtubule-associated protein 1 light chain 3-II (LC3-II) and sequestosome-1 (SQSTM1/p62). These results reveal that the RSV-SLNs may have great potential used as a novel anti-fatigue sports nutritional supplement.

## 1. Introduction

Fatigue is defined as the difficulty in initiating or sustaining voluntary activities, and can be classified into mental and physical fatigue. Physical fatigue is thought to be accompanied by deterioration in physical performance. Among the fatigue mechanisms, the “radical theory” has been attracted more attention [1]. The classical “radical theory” suggests that intense exercise can produce an imbalance between the body’s oxidation system and its anti-oxidation system. The accumulation of reactive free radicals will lead to oxidative stress which can induce injury to tissues or organs and is also one of the reasons for fatigue [2,3]. Fatigue is a physiological phenomenon that arises from physical stress or exhaustive exercises leading to the reduction of physical endurance capacity [4]. Some reports showed that supplementing with exogenous antioxidants has been considered to have an important role in decreasing the degree of exercise-induced oxidative stress and improving the physiological condition [5].

Resveratrol (trans-3, 5, 4′-trihydroxystilbene) is a natural polyphenolic phytoalexin which can be found in various common foods like grapes, peanuts, various berries, and red wine. It offers various of health benefits, such as antioxidant, anti-inflammatory, anticancer activity, anti-obesity, anti-aging, and cardiovascular protective effects. Some reports showed that the antioxidant effect of resveratrol in many diseases are related to the protection against oxidative damage and reduction of the oxidative stress [6,7,8]. However, its pharmacokinetic properties are not so favorable due to its poor aqueous solubility, rapid and extensive metabolism and poor bioavailability, which limit its immense potential. Therefore, in order to overcome these problems, it is necessary to exploit the nanosystem as an efficient delivery to protect resveratrol from degradation and improve its clinical therapy.

In recent years, the development of drug delivery system has attracted more attention. Solid lipid nanoparticles (SLNs) have been reported to have potential for oral delivery of lipophilic drugs owing to a number of beneficial characteristics: SLNs are nontoxic and biocompatible with high payload capacity, and it can raise the solubility and stability of loaded drugs [9,10,11,12]. SLNs are being looked upon as a novel method for improving the pharmacokinetic properties, as well as enhancing targetability and bioavailability of resveratrol. The major determinants of bioavailability after oral administration of small drug molecule include the aqueous solubility, membrane permeability and metabolic stability of the given drug [13,14,15].

In our previous research, we successfully loaded resveratrol into SLNs for improving mitochondrial quality control [16]. In this study, the resveratrol-loaded solid lipid nanoparticles (RSV- SLNs) were designed as a nano-antioxidant against the physical fatigue. Particle size, TEM, zeta potential, differential scanning calorimetry and Raman spectroscopy pattern of RSV-SLNs were investigated. Furthermore, the in vivo anti-fatigue effect and the molecular mechanisms of antioxidant by RSV-SLNs supplementation through the analysis of silent information regulator 2 homolog 1 (SIRT1) protein expression.

## 2. Results

### 2.1. Physicochemical Characteristics of RSV-SLNs

The TEM images (Figure 1A,B) showed that RSV-SLNs and SLNs were both spherical with smooth surfaces morphology. The length of the discrete spheres was in the range of 50–70 nm. Analysis of homogeneously dispersed RSV-SLNs (Figure 1C) showed a narrow particle size distribution. The equivalent mean hydrodynamic diameter was about 156.10 nm and most particles were distributed within a range of 120–170 nm, and the suspension was stable. Meanwhile, the SLNs revealed a particle size distribution with an average diameter 78.30 nm (Figure 1E), and exhibited significant differences in particle size when compared with RSV-SLNs.

### 2.2. Differential Scanning Calorimetry (DSC) Measurements

DSC analysis was conducted to investigate the melting and crystallization behavior of both SLNs and RSV-SLNs. The thermograms curves of physical complex SLNs, SLNs, physical complex RSV-SLNs and RSV-SLNs were shown in Figure 2. The melting process for physical complex SLNs took place with maximum peaks at 51.15 °C and 73.15 °C. The melting process for SLNs took place with maximum peaks at 69.60 °C (Figure 2A). The melting process for physical complex RSV-SLNs took place with maximum peaks at 50.157 °C and 75.15 °C. The melting process for RSV-SLNs took place with maximum peaks at 74.10 °C and 254.60 °C. (Figure 2B).

### 2.3. Raman Spectroscopy

Raman spectroscopy was used to obtain information about the structure and properties of molecules from their vibrational transitions. Previous report has been shown that Raman spectra of lipids are sensitive to conformational, packing and dynamical changes involving hydrocarbon chains. In this study Raman spectroscopy was used to investigate the conformational order of hydrocarbon chains, i.e., if and how oil loading changes the lipid chain arrangement. Raman spectroscopy was a particularly useful technique as it involved no sample preparation and most importantly allowed measurements in the presence of water [17,18].

Raman spectra were obtained from RSV, SLNs and RSV-SLNs (Figure 3). Both prominent peaks at 1631 cm^−1^ and 1604 cm^−1^ (Figure 3C) of resveratrol were observed which assigned to a combination of ν(C=C) and δ(C–H) vibrations of the trans-olefin carbons together with ν(C=C) vibrations of the phenyl rings. A typical room temperature Raman spectrum of silica aerogel was shown (Figure 3B). For the SLN spectra, the characteristic peaks at 2881 cm^−1^, 2845 cm^−1^, 1298 cm^−1^, 1128 cm^−1^ and 1062 cm^−1^ were observed which source from the cetyl palmitate. However, the characteristic MCT peak at 1743 cm^−1^ was not detectable. The Raman spectrum of the nanoemulsion is characterized by random coiled chains, while those of SLN-formulations clearly show sharp bands at 1128 cm^−1^ and 1062 cm^−1^ indicating the high conformational order of the acyl chains. In the spectrum of RSV-SLNs, four characteristic peaks weak intensities peaks nearly 1631 cm^−1^ and 1604 cm^−1^, 1128 cm^−1^ and 1062 cm^−1^ from both RSV and SLN were visible.

### 2.4. Effect of RSV-SLNs on Exercise Fatigue Related Tests

The running training test was used for evaluating endurance capacity. A direct measure of an anti-fatigue effect was the increase in exercise tolerance. We investigated whether RSV-SLNs could improve exercise performance by measuring exhausted time and running distance. As shown in Figure 4, the exhaustive exercise (EE) duration and distance of EE + RSV-SLNs group were significantly longer than that of EE group (*p* < 0.05). The results indicated that RSV-SLNs might possess an anti-fatigue effect.

### 2.5. The Biochemical Parameters of Blood Related to Fatigue

In the Table 1, we found that the serum markers of aspartate aminotransferase (AST) and alanine aminotransferase (ALT), liver-related parameters, were increased significantly (*p* < 0.05 respectively), indicating that liver was damaged as a consequence of exhaustive exercise. Furthermore, the serum AST levels in the EE + RSV-SLNs group significantly decreased by 25.1% (*p* < 0.05), and the serum ALT levels in the EE + RSV and EE + RSV-SLNs groups were significantly lower than that of the EE group by 32.0% (*p* < 0.05) and 40.2% (*p* < 0.01), respectively.

### 2.6. Effect of RSV-SLNs Supplementation on the Muscular Morphology of Mice that Underwent Excessive Endurance Exercise

Muscular tissue injury was assessed on the basis of the morphological changes detected by hematoxylin-eosin (HE) staining (Figure 5A–D). The cross-section of the gastrocnemius (GAS) muscle in the SC group displayed normal morphology, that is, the fibers had peripheral nuclei and lacked lesions and inflammatory processes (Figure 5A). Moreover, the EE group exhibited intense muscular fiber injury with disorganized muscle nuclear orientation, edema, neutrophil infiltration, and myonecrosis (Figure 5B). The degree of the alleviation of muscular damage was better in the EE + RSV group than in the EE group (Figure 5C). The characteristics of the centralized nuclei were observed, and they represented newly regenerated skeletal muscles. Muscular fiber injury was significantly reduced in the EE + RSV-SLNs group, with reduced inflammatory infiltration, edema, and myonecrosis, and increased presence of normal fiber shape (Figure 5D).

### 2.7. Antioxidative Effect of RSV-SLNs on Liver

The exhaustive exercise-induced lipid peroxidation in mice was estimated by measuring the levels of malondialdehyde (MDA). As shown in Figure 6, the liver MDA levels of the EE + RSV-SLNs group was significantly lower than those of the EE group (*p* < 0.05). Superoxide dismutase (SOD) dismutase superoxide radicals to form H_2_O_2_ and O_2_. GSH-PX is an enzyme responsible for reducing H_2_O_2_ or organic hydroperoxides to water and alcohol. As shown in Figure 6, compared with the EE group, the SOD levels in liver of EE+ RSV-SLNs group was significantly higher (*p* < 0.01).the liver glutathione peroxidase (GSH-PX) levels of the EE+ RSV-SLNs group was significantly higher than those of the EE + RSV group (*p* < 0.01).Lipid peroxidation appears to be an important mechanism underlying exercise-induced muscle damage [19].

### 2.8. Effect of RSV-SLNs Supplementation on Lipid Peroxidation and Antioxidant Status in Mice Subjected to Excessive Endurance Exercise

Oxidative stress is a condition in which the production and clearance of free radicals are unbalanced, and their clearance rate is insufficient to mediate their production, such as during exhaustive exercise. To maintain the balance between production and clearance of free radicals, the body’s antioxidant defense system needs to be activated by antioxidant supplementation, which is a key role in regulating cellular functions. In Figure 7, the MDA activity in the skeletal muscle significantly increased (*p* < 0.05), and SOD activity decreased (*p* < 0.05). The catalase (CAT) activity mildly decreased in response to excessive endurance exercise. The activities of SOD and antioxidant enzymes were upregulated after RSV-SLNs application (*p* < 0.05), and CAT was upregulated in the skeletal muscle of the mice subjected to endurance exercise after RSV (*p* < 0.05) or RSV-SLNs (*p* < 0.05) treatment was administered.

### 2.9. Effect of Res-SLNs on the Expression of SIRT1, LC3-II and p62 Signaling

The study then focused on the molecular mechanisms underlying the enhancement of anti-inflammation/antioxidant by RSV-SLNs supplementation through the analysis of SIRT1 protein expression. The Western blot analysis showed that long-term endurance exercise resulted in an increase in the SIRT1 protein expression (*p* < 0.05); however, the increased expression of SIRT1 reversed after RSV or RSV-SLNs supplementation (*p* < 0.05), which suggested that the RSV-SLNs might promote anti-inflammatory and antioxidant activity of RSV via downregulation of SIRT1 in response to excessive endurance exercise (Figure 8A).

Furthermore, the expression levels of microtubule-associated protein 1 light chain 3-II (LC3-II) and sequestosome-1 (SQSTM1/p62) were investigated to verify RSV-SLNs regulates accumulation of SIRT1 via the autophagic pathway. The Western blot analysis showed that the LC3-II and p62 protein expression levels significantly decreased after long-term endurance exercise (*p* < 0.05), whereas the increase in the levels of LC3-II and p62 protein expression was determined after RSV-SLNs administration (*p* < 0.05) (Figure 8B).

## 3. Discussion

### 3.1. Characteristics of RSV-SLNs

Resveratrol-loaded SLNs were prepared using a simple, economical and reproducible method called emulsion evaporation-solidification at low temperature method. From the TEM images (Figure 1) the length of the discrete spheres was in the range of 50–70 nm which consist with previous reports [20]. RSV-SLNs and SLNs were both spherical with smooth surfaces morphology. Zeta potential is an important and useful tool to indicate particle surface charge, which could be used to predict and control the stability of colloidal suspensions. The charged particles will repel each other if the systems have high positive or negative values of zeta potential, and thus overcoming the natural tendency to aggregate. The RSV-SLNs and SLNs had an average zeta potential of −23.6 mV and −23.6 mV respectively in suspension (Figure 1D,F). The negative potential is high enough to avoid the aggregation of nanoparticles and keep their dispersion system stable.

DSC enables an insight into the melting and recrystallization behavior of crystalline materials, such as lipid nanoparticles. The breakdown of the crystal lattice by heating reveals further information on the polymorphism, crystal ordering, eutectic mixtures, and/or the glass transition process. The RSV-SLNs thermogram showed a sharp endothermic peak at 254.6 °C, corresponding to the resveratrol melting point and reflecting its high crystallinity. There was no endothermic peak for resveratrol was obtained from physical complex RSV-SLNs, clearly indicating that resveratrol loaded in the SLNs existed as a crystalline form not an amorphous state. This was in contrast to the work of previous research which found resveratrol loaded in the SLNs existed as an amorphous form [21].

From the Raman spectrum of three samples in Figure 3, the spectrum of RSV-SLNs contained all the characteristic peaks in both resveratrol and SLN that demonstrate resveratrol existed in the RSV-SLNs. Compared with RSV crystal, all peaks of RSV-SLNs were detected at the same position, which indicated that the drug remained intact in the preparation process and no additional characteristic peaks appeared [22,23].

### 3.2. The Anti-Fatigue Related Effect of RSV-SLNs

A previous study reported that RSV provided a protective effect against chemical-induced oxidative stress on AST, ALT, and lactate dehydrogenase (LDH) activities, and blood urea nitrogen (BUN) and creatinine (CRE) levels, which is consistent with our study [24]. Additionally, the beneficial effect of RSV-SLNs on exhaustive exercise-induced hepatic injury is better than that of RSV. As to kidney function, the biochemistry of BUN, and CRE can reflect renal damage. Our data showed that CRE was significantly decreased in the EE + RSV-SLNs group by 27.1%, and BUN in the EE + RSV and EE + RSV-SLNs groups were significantly lower by 20.6% and 30.8%, respectively, compared to values of EE group. Above all, RSV-SLNs may possibly have potential applications for liver and renal protection due to its antioxidant activity.

Lipid peroxidation represents oxidative tissue damage caused by hydrogen peroxide, superoxide anions and hydroxyl radicals, resulting in structural alteration of the membrane, release of cell and organelle content and loss of essential fatty acids with formation of cytosolic aldehyde and peroxide products [23]. The MDA is a metabolite of phospholipid peroxidation, is a popular index of first condition on living body oxidative damage [25]. In this study, the data showed that RSV-SLNs significantly decreased MDA levels of mice, which indicated that RSV-SLNs could reduce lipid peroxidation and attenuate exhaustive exercise-induced oxidative damage.

There is an evidence that ROS exceeds the normal physiological coping range during exhaustive exercise, accumulation of ROS and decrease in antioxidant status could be resulted in fatigue. This condition increases oxidative stress and leads to lipid peroxidation and oxidative modifications of proteins and DNA [26]. The betterment of antioxidant enzyme activities can ameliorate fatigue because the antioxidant defense gets weaker during exhaustive exercise-induced fatigue. Muscle and liver cells contain endogenous cellular defense mechanisms to eliminate reactive oxygen species. The antioxidant enzymes such as SOD and GSH-PX have an important function in lighting the toxic effects of ROS, and the improvement in the antioxidant enzyme activities can help to fight and relieve physical fatigue [27,28]. In this study, the data showed that RSV- SLNs increased SOD and GSH-PX of mice significantly. We speculate that RSV- SLNs would improve the antioxidant defense of the mice after extensive exercise and confer anti-fatigue capability.

Combining with the results of lipid peroxidation and antioxidant status in mice subjected to excessive endurance exercise, these results indicated that RSV-SLNs supplementation not only improved endurance performance but also restored the imbalance between oxidants and antioxidants.

### 3.3. The Molecular Mechanisms of RSV-SLNs

The results of expression of SIRT1, LC3-II and p62 corresponded to the changes of SIRT1 expression levels, suggesting that the autophagy pathway was activated by RSV-SLNs-induced SIRT1. It is reported that RSV enhanced autophagic flux through the SIRT1 axis [29,30,31]. Therefore, SIRT1 might play an important role in the regulation of autophagy and autophagic flux. The purpose of this study was to explore whether RSV-SLNs supplementation protects skeletal muscle from excessive endurance exercise-induced inflammatory and oxidant injury through enhancing autophagic flux dependent on SIRT1 and further investigate the possible mechanism of RSV-SLNs protective effects. The results showed that long-term endurance exercise increased the expression of SIRT1 in the skeletal muscle tissues, and RSV-SLNs supplementation restored the expression of SIRT1. Meanwhile, the effect of excessive endurance exercise on the expression of LC3-II and p62 was abolished when subjected to RSV-SLNs supplementation. The protective effects of RSV-SLNs, which included downregulating the expression of SIRT1 and increasing LC3-II and p62, demonstrated that RSV-SLNs is essential for the amelioration of long-term endurance exercise-impaired autophagic flux in the skeletal muscle by downregulating SIRT1.

## 4. Materials and Methods

### 4.1. Materials

Stearic acid, leclthin, ethyl alcohol absolute, dibasic sodium phosphate, sodium dihydrogen phosphate, sodium phosphotungstate and trichloromethane were purchased from China National Medicine Group, Shanghai Chemical Reagents Company (Shanghai, People’s Republic of China), and were used without further purification. Resveratrol in the purity of 99% were obtained from Aladdin Industrial Corporation (Shanghai, People’s Republic of China). Myrj52 and acetonitrile were purchased from Sigma-Aldrich Co. (St Louis, MO, USA). Deionized water was decarbonated by boiling before use in all applications.

### 4.2. Animals

Male C57BL/6J mice were purchased from SLAC Laboratory Animal Research Center (Shanghai, China) and housed individually in a controlled environment (12/12 h light/dark cycle, 08:00 h to 20:00 h, humidity: 60% ± 5%, temperature: 23 ± 2 °C). All mice were provided with free access to purified water and food throughout the experiment and weighed weekly. After acclimatization for 1 week, all mice were randomly divided into four groups with six mice each: (1) sedentary control group (SC); (2) exhaustive exercise (EE); (3) exhaustive exercise combined resveratrol supplementation (EE + RSV); and (4) exhaustive exercise combined with RSV-SLNs supplementation (EE + RSV-SLNs). The SC group and EE group were administered distilled water at 10 mg/kg of body weight. The amount of resveratrol was adjusted to the equal to that of resveratrol-SLN treatment group. All treatments were administered once per day, 6 days/week and lasted for 8 weeks. The forced running test was conducted on the last day and corresponding biochemical parameters were measured.

### 4.3. Preparation of Resveratrol-Solid Lipid Nanoparticles (RSV-SLNs)

Resveratrol-SLN was prepared through emulsification and low-temperature solidification method. 150 mg resveratrol, 200 mg Stearic acid, 100 mg leclthin ultrasonic, and 10 mL trichloromethane were added into a 25 mL pear-shaped flask and dissolved by ultrasound. This was the organic phase. 250 mg Myrj52 was dissolved in 30 mL of high-purity water and this was the aqueous phase. The organic phase was then injected into the aqueous phase and the mixture was stirred at 1000 rpm at (75 ± 2) °C. The stirring lasted for approximately 2.5 h, until the organic solvent completely disappeared and the system volume condensed to about 5 mL. The semitransparent system was then quickly added to another 10 mL of cold water and stirred at 1000 rpm, under the temperature of 0–2 °C. The stirring lasted for 2 h. The product was denoted as resveratrol-SLN colloidal suspension mixed liquid. The resultant precipitates were centrifuged (4 °C) at 15,000 rpm for 60 min and washed twice with double distilled water to remove excess of surfactant and untrapped drug. SLN were then freeze-dried at −80 °C refrigerator for 24 h.

### 4.4. Morphology Detected by Transmission Electron Micrographs (TEM), Photon Correlation Spectroscopy (PCS) and Zeta Potential Measurement

Five microliters of diluted solution of sample were placed on carbon-formvar coated 400 mesh spacing grids and left to adsorb for 5 min. Negative staining was performed with 2% filtered aqueous solution of uranyl acetate for 45 s. The grids were visualized using a JEM 1400 electron microscope (JEOL-1230, Tokyo, Japan) at 80 kV.

The average particle size (z-average size) and size distribution were measured using photon correlation spectroscopy (PCS) (LS230; Beckman Coulter) at 25 °C under a fixed angle of 90° in disposable polystyrene cuvettes. The measurements were recorded using a He–Ne laser of 633 nm. Zeta potential distribution and polydispersity index of the nanoparticles were analyzed by Nano ZS (Malvern Instruments, Malvern, UK). The zeta potential of prepared SLN was measured to assess the surface charge and stability. Samples were prepared by re-dispersing the lyophilized nanoparticle in double distilled water as dispersing medium (dielectric constant 78.5, viscosity 0.8872 cP, refractive index 1.330). For each sample the measurements were repeated thrice.

### 4.5. Differential Scanning Calorimetry (DSC) Analysis

A differential scanning calorimetry (DSC) examination was performed on a DSC822e DSC (Mettler Toledo). A heating rate of 10 °C min^−1^ was employed in the temperature range 25–400 °C. An empty aluminum pan was used as reference standard. Analysis was carried out under nitrogen purge at 50 mL min^−1^.

### 4.6. Raman Spectra Analysis

Raman spectroscopy LabRam HR 800 (Horiba Jobin Yvon, Edison, NJ, USA) was used in the present study, which were recorded from 0 to 3000 cm^−1^ using a 633 nm wavelength laser (source power 20 mW). This spectrometer is equipped with a confocal microscope (Olympus BX40), a piezoelectric x, y stage, and a CCD detector.

### 4.7. In Vivo Anti-Fatigue Effect of RSV-SLNs

#### 4.7.1. Effect on Running Time to Fatigue

The whole running training was conducted using a motor treadmill (Hangzhou, China). After acclimating for 5 days, mice were exercised at intensive intensity for 2 weeks (speed at 30 m/min at a 5% grade, 120 min per day), and then were exercised until volitional fatigue (e.g., failure to maintain pace with the treadmill) at gradually increasing speeds from 20 to 40 m/min at a 5% grade for another 2 weeks. The exhaustive distance of mice in each group were recorded. Mice ran without electric shock or prodding. Mice in the control group were exposed to noise and handling, similar to other groups to regulate stress associated with the exercise. The running time to exhaustion was used as the index of the forced running capacity.

#### 4.7.2. The Biochemical Parameters of Blood and Tissue Sampling

At the end of their respective manipulation, all mice were fasted overnight and anesthetized with 2% intraperitoneal injection of sodium pentobarbital (6.5 mg/100 g weight). Blood samples were collected from tail veins and centrifuged at 1100× *g* for 10 min to separate serum and stored at −20 °C until analysis. Then, the content of BUN, creatine kinase (CK), LDH, AST and ALT were determined. The liver was collected to be made into 10% homogenates with saline as soon as possible. The homogenized liver tissues were centrifuged at 2500 rpm for 10 min to get the supernatant for following test. MDA content, SOD activity and GSH-Px activity were determined with commercial assay kits (Nanjing Jiancheng Bioengineering Institute, Nanjing, China) following the manufacturer’s instructions.

#### 4.7.3. Blood and Tissue Sampling

At the end of each manipulation, all the mice were subjected to fasting overnight and anesthetized by intraperitoneally injecting 2% sodium pentobarbital (6.5 mg/100 g body weight). Blood samples were collected from tail veins and centrifuged at 1100× *g* for 10 min. Serum samples were separated and stored at −20 °C for analysis. Afterward, the gastrocnemius muscle tissues were completely excised and weighed individually. A portion of the GAS muscle tissue was kept for HE staining and TEM or homogenized immediately to determine oxidative stress biomarkers and mitochondrial respiratory chain enzymes. The remaining portions were stored at −80 °C until further analysis.

### 4.8. Effect of Res-SLNs on the Expression of SIRT1, LC3-II and p62 Signaling

Total proteins were extracted from frozen gastrocnemius muscle tissues, and protein content was determined by BCA assay (Thermo Scientific, Waltham, MA, USA). Loading proteins were separated by SDS-PAGE (10% and 12% resolving gel), transferred onto polyvinylidene difluoride membrane (Millipore Corp., Bedford, MA, USA), and blocked with nonfat milk. The membranes were incubated with primary antibodies overnight at 4 °C. The following primary antibodies were used: anti- SIRT1 (1:1000; #5114; cell signaling), anti- LC3-II (1:1000; #4108; cell signaling), anti- p62 (1:1000; #04-1557; Millipore), and glyceraldehyde phosphate dehydrogenase (GAPDH; 1:5000; AB2000; Abways technology). After incubation with peroxidase-conjugated secondary antibodies (1:2000; SC-2054; Santa Cruz) for 1 h, the membranes were rinsed three times, and then the protein bands were visualized with an ECL Plus Kit (Amersham, Stockholm, Sweden) and quantified using Image J software (Image Processing and Analysis in Java, Bethesda, MD, USA).

### 4.9. Statistical Analysis

For statistical analysis, the data were presented as the mean ± standard deviation of three independent experiments. *p* values were calculated by one-way ANOVA.

## 5. Conclusions

In this research, resveratrol had been successfully incorporated into solid lipid nanoparticle emulsification and low-temperature solidification method. The composition, structure, particle shape and the size were characterized by a series of assays. The in vivo anti-fatigue effect showed that RSV-SLNs have the protective effect against chemical-induced oxidative stress on AST, ALT, and LDH activities, and BUN and CRE levels compared to RSV groups and control groups. RSV-SLNs could also reduce lipid peroxidation and attenuate exhaustive exercise-induced oxidative damage. Besides, RSV-SLNs is essential for the amelioration of long-term endurance exercise-impaired autophagic flux in the skeletal muscle by downregulating SIRT1. These results indicated that the SLN can serve as a better carrier for resveratrol to increase the endurance capacity and facilitate recovery from fatigue, which may use as a novel anti-fatigue sports nutritional supplements in the future.

## Figures and Tables

**Figure 1 molecules-25-05302-f001:**
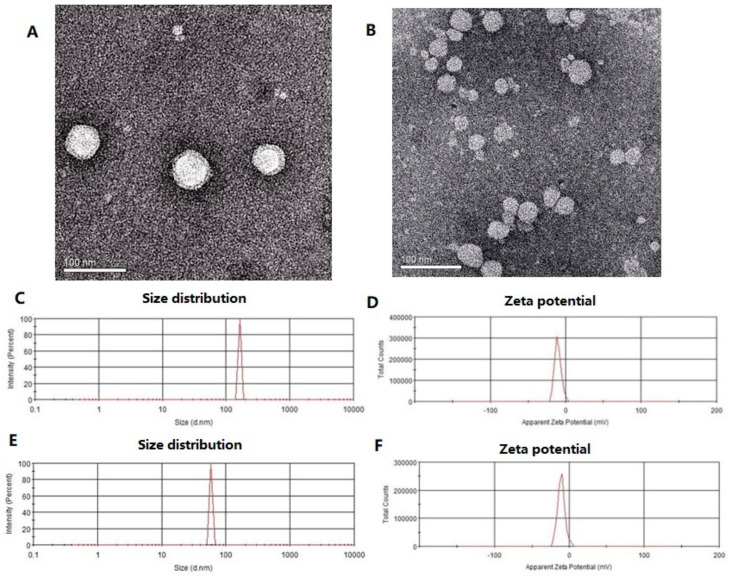
Morphology of RSV-SLNs and SLNs observed by TEM (**A**,**B**); The size distribution and zeta potential for RSV-SLNs (**C**,**D**); The size distribution and zeta potential for SLNs (**E**,**F**). RSV-SLNs: resveratrol-loaded solid lipid nanoparticles; SLNs: solid lipid nanoparticles; TEM: transmission electron microscopy.

**Figure 2 molecules-25-05302-f002:**
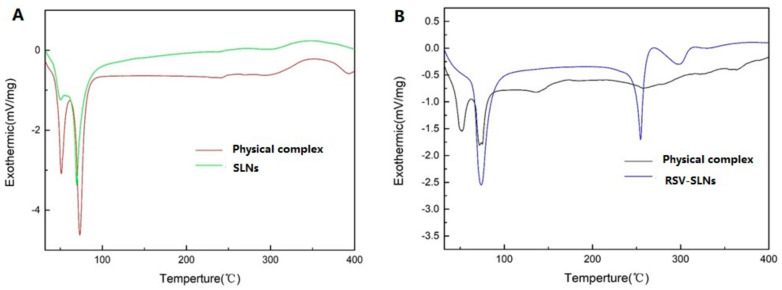
DSC heating curves of physical complex SLNs and SLNs (**A**); physical complex RSV-SLNs and RSV-SLNs (**B**) from 0 to 400 °C.

**Figure 3 molecules-25-05302-f003:**
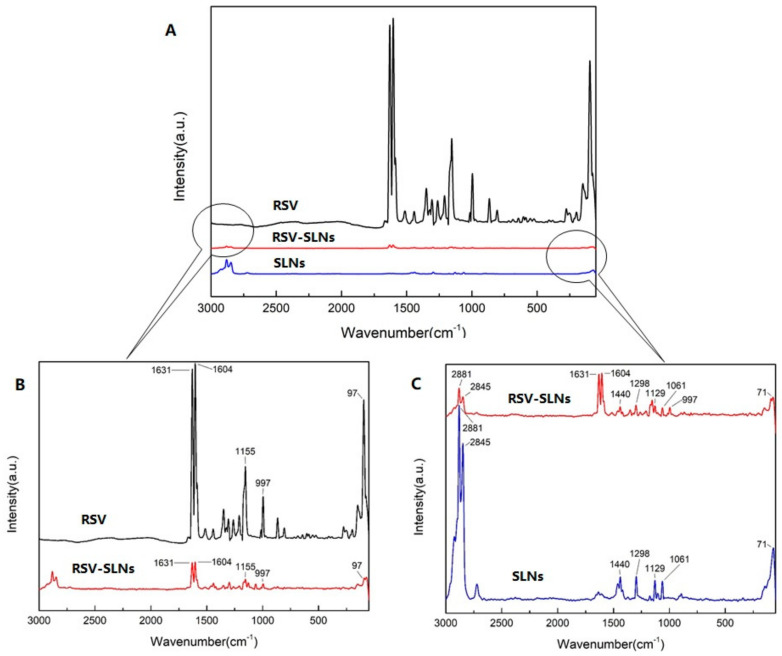
Raman spectra of RSV, RSV-SLNs and SLNs (**A**); the larger views of RSV, RSV-SLNs (**B**) and RSV-SLNs, SLNs (**C**).

**Figure 4 molecules-25-05302-f004:**
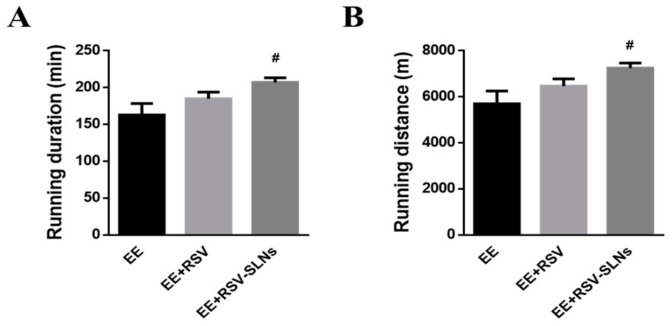
Effect of RSV-SLNs on running exhausted time (**A**) and running distance (**B**) of mice. Data are expressed as mean ± SD of six mice in each group. ^#^
*p* < 0.05 compared with EE group. EE: exhaustive exercise group; EE + RSV: exhaustive exercise combined with resveratrol supplementation; EE + RSV-SLNs: exhaustive exercise combined with RSV-SLNs supplementation.

**Figure 5 molecules-25-05302-f005:**
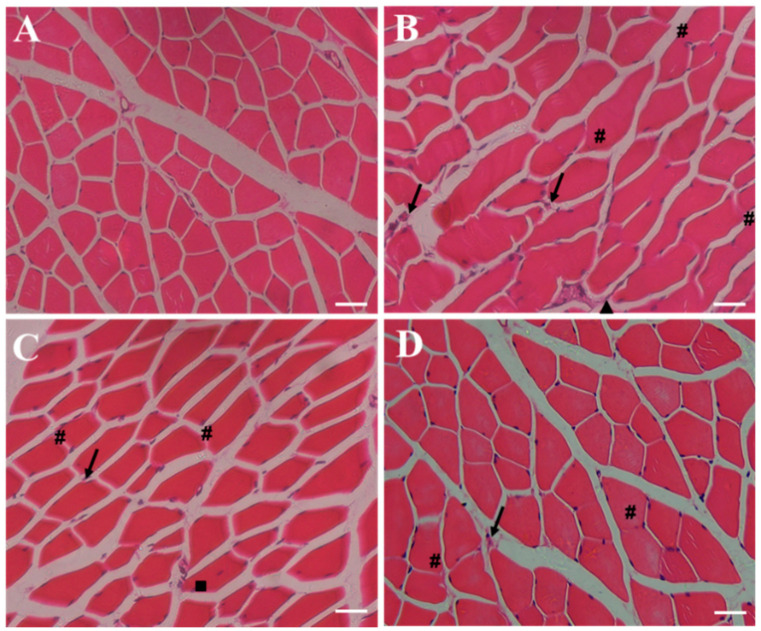
RSV-SLNs relieves skeletal muscle injury induced by excessive endurance exercise. Cross-sections of the histological photomicrography of GAS muscles with HE staining from the four groups of mice. (**A**): sedentary control (SC); (**B**): exhaustive exercise group (EE); (**C**): exhaustive exercise combined with resveratrol supplementation (EE + RSV); (**D**): exhaustive exercise combined with RSV-SLNs supplementation (EE + RSV-SLNs). The black arrow denotes inflammatory cell infiltrate; triangle refers to myonecrosis; square corresponds to centrally nucleated fibers; and pound sign indicates edema. The scale bars represent 200 µm in skeletal muscle.

**Figure 6 molecules-25-05302-f006:**
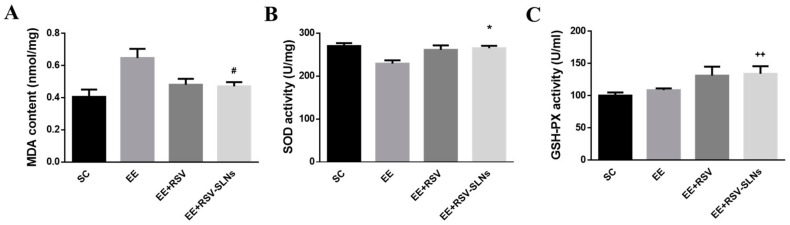
Effect of RSV-SLNs on liver biochemical variables of mice after treadmill exhausted test. The levels of MDA content (**A**), SOD activity (**B**), and GSH-PX activity (**C**) were detected in the skeletal muscle from four groups: sedentary control (SC), exhaustive exercise group (EE), exhaustive exercise combined with resveratrol supplementation (EE + RSV), exhaustive exercise combined with RSV-SLNs supplementation (EE + RSV-SLNs). ^#^
*p* < 0.05 compared with EE group. * *p* < 0.05 compared with SC group. ^++^
*p* < 0.01 compared with EE + RSV group. MDA: malondialdehyde; SOD: superoxide dismutase; GSH-PX: glutathione peroxidase.

**Figure 7 molecules-25-05302-f007:**
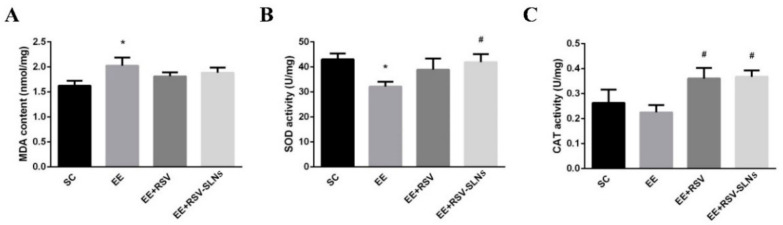
RSV-SLNs attenuates lipid pro-oxidant status during excessive endurance exercise. (**A**) MDA content. Antioxidant enzyme levels of (**B**) SOD and (**C**) CAT in the skeletal muscle from four groups of mice: sedentary control (SC), exhaustive exercise group (EE), exhaustive exercise combined with resveratrol supplementation (EE + RSV), exhaustive exercise combined with RSV-SLNs supplementation (EE + RSV-SLNs). * (*p* < 0.05) vs. SC group; ^#^ (*p* < 0.05) vs. EE group. MDA: malondialdehyde; SOD: superoxide dismutase; CAT: catalase.

**Figure 8 molecules-25-05302-f008:**
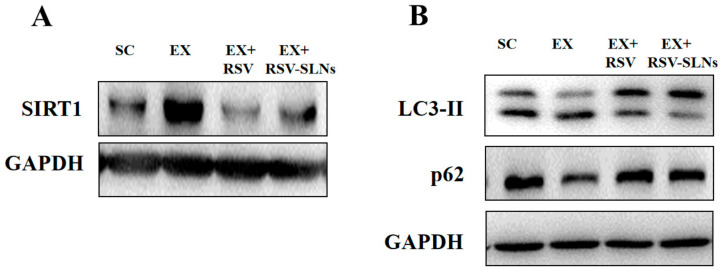
RSV-SLNs regulates SIRT1 and LC3-II/p62 pathway. (**A**) The expression levels of SIRT1 protein was assayed with Western blot analysis. (**B**) The expression levels of LC3-II and p62 protein were detected by Western blot.

**Table 1 molecules-25-05302-t001:** Biochemical analysis of the RSV-SLNs supplementation in exhaustively exercised mice.

Parameter	SC	EE	EE + RSV	EE + RSV-SLNs
AST (U/L)	123.70 ± 6.60	178.50 ± 8.30 *	155.90 ± 9.30	133.70 ± 6.60 ^#^
ALT (U/L)	42.90 ± 7.80	57.20 ± 2.70 *	38.90 ± 4.20	34.20 ± 1.80 ^##^
ALP (U/L)	152.50 ± 4.27	140.90 ± 5.31	121.97 ± 16.20	90.75 ± 8.05 *
LDH (U/L)	409.90 ± 78.00	445.20 ± 150.40	372.60 ± 95.40	489.40 ± 43.80
ALB (g/L)	23.23 ± 0.93	24.60 ± 0.16	24.53 ± 1.25	24.90 ± 0.30
T-BIL (µmol/L)	3.47 ± 0.58	3.23 ± 0.54	4.23 ± 1.10	4.45 ± 0.65
TP (g/L)	51.23± 2.16	56.73 ± 2.89	60.83 ± 6.45	66.70 ± 8.40
BUN (mmol/L)	12.33 ± 0.72	13.60 ± 0.78	10.80 ± 0.56 ^#^	9.41± 0.72 ^##^
CRE (µmol/L)	7.19 ± 1.13	6.58 ± 0.54	6.13 ± 0.63	4.80 ± 0.89 *^,#^

Serum biochemical indicators from four groups of mice: sedentary control (SC), exhaustive exercise group (EE), exhaustive exercise combined with resveratrol supplementation (EE + RSV), exhaustive exercise combined with RSV-SLNs supplementation (EE + RSV-SLNs). * *p* < 0.05, and ** *p* < 0.01 vs. SC group; ^#^
*p* < 0.05, and ^##^
*p* < 0.01 vs. EE group. AST: aspartate aminotransferase; ALT: alanine aminotransferase; ALP: alkaline phosphatase; LDH: lactate dehydrogenase; ALB: albumin; T-BIL: total bilirubin; TP: total protein; BUN: blood urea nitrogen; CRE: creatinine.

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
