# Peer review of "In Vivo Effect of Resveratrol-Loaded Solid Lipid Nanoparticles to Relieve Physical Fatigue for Sports Nutrition Supplements"

_molecules, 2020, doi:10.3390/molecules25225302_

Round 1

Reviewer 1 Report

The paper “In vivo anti-fatigue Effect of Resveratrol- Loaded Solid lipid Nanoparticles for Sports Nutrition Supplement” is interesting and  acceptably written but, in my opinion, few points should be improved before the publication because it turns out to be a bit confusing.  

- I would recommend to add “physical” in the title to indicate that the study is about physical and not mental “anti-fatigue effect”

- Lines 113-115: Explain how the AST, ALT, ALP, CRE, BUN, 114 MDA contents and SOD, GSH-PX parameters are responsible for the mechanism behind the anti-fatigue effect.

- The results are shown with different decimals. Standardize all data with the same decimals along the text (for example Table 1).

- Figure 6: Indicate the unit of measure on the y axis of Figure 6B; detail the legend of the figure explaining in detail what the images 6a, 6b, and 6c represent.

- Explain the acronyms of the analyzed samples in the legends of all the images.

Minor revisions:

- Explain all acronyms the first time you use it in the text, for example:

SLNs, SIRT1, LC3-II and p62 in the abstract;

            “EE” in the image legend of Figure 4 at line 107

            “NA” in Table 1 at line 125

            “HE” at line 130

            “MDA” at line 147

- Missing space: Line 36 after the point “[2-3].Fatigue”;

                         Line 118 after 0.05 “p < 0.05respectively”

                           Line 220 aftere RSV “RSV.As”

- Double space: Line 59 “control [16]”.

                        Line 150 “alcohol. As”

- Should be changed in all text:

          “in vivo” with “in vivo” in italics

          “vs.” with “vs” in italics

           “ml” with “mL” lines 287, 288, 292

Author Response

Dear Reviewer,

  Thank you for your valuable and thoughtful comments on our manuscript. According to the comments, we have carefully revised our paper. Some alterations have been made and some more details have been added in the revised form. The corrections of the manuscript are highlighted in red through the text. All pages and lines are numbered consecutively. The corrections and responses to the comments are listed as follows, or please see the attachment:

  1. I would recommend to add “physical” in the title to indicate that the study is about physical and not mental “anti-fatigue effect”

Response:Thanks for the valuable and thoughtful suggestions. We have revised the title as “In vivo Effect of Resveratrol-Loaded Solid lipid Nanoparticles to Relieve Physical Fatigue for Sports Nutrition Supplements.”

  1. Lines 113-115: Explain how the AST, ALT, ALP, CRE, BUN, 114 MDA contents and SOD, GSH-PX parameters are responsible for the mechanism behind the anti-fatigue effect.

Response:Thanks for the valuable and thoughtful suggestions. Growing studies indicate that reactive oxygen species are responsible for exercise-induced protein oxidation, and contribute strongly to muscle fatigue [1]. Muscle and liver cells contain endogenous cellular defense mechanisms to eliminate reactive oxygen species. In addition, lipid peroxidation was frequently used as an indication of tissue oxidative stress as a result of a free radical attacking the cell membrane. The primary antioxidant enzymes include SOD, GSH-Px and CAT [2]. SOD eliminate superoxide radicals to form H2O2 and O2. GPH-Px is an enzyme responsible for reducing H2O2 or organic hydroperoxides to water and alcohol, respectively. CAT catalyses the breakdown of H2O2 to form water and O2. These antioxidant defense mechanisms become weaker during chronic fatigue and other disease conditions [3]. MDA is the metabolism products due to lipid peroxidation after exercise. The lower production of MDA shows the reducing lipid peroxide and suppressing tissue damage in response to exercise. So, the improvement in the activities of these defense mechanisms can help to fight and relieve physical fatigue.

  Besides, a previous study reported that RSV provided a protective effect against chemical-induced oxidative stress on blood parameters, such as, AST, ALT, LDH activities, and BUN, CRE levels [4], which is consistent with our study. The results in our study showed that the beneficial effect of RSV-SLNs on exhaustive exercise-induced hepatic injury was better than that of RSV. As to kidney function, the biochemistry of BUN, and CRE can reflect renal damage. Our data showed that CRE was significantly decreased in the EE+RSV-SLNs group by 27.1%, and BUN in the EE+RSV and EE+RSV-SLNs groups were significantly lower by 20.6% and 30.8%, respectively, compared to the values of EE group. Above all, RSV-SLNs may possibly have potential applications for liver and renal protection due to its antioxidant activity.

  According to the reviewer’s valuable suggestions, we have supplemented the detailed explanations about it in the discussion part. Please check the Line 236-244 and Line 255-260 in the revised manuscript.

[1] Powers, S. K., DeRuisseau, K. C., Quindry, J., & Hamilton, K. L. (2004). Dietary antioxidants and exercise. Journal of Sports Sciences, 22, 81–94.

[2] Tharakan, B., Dhanasekaran, M., & Manyam, B. V. (2005). Antioxidant and DNA protecting properties of anti-fatigue herb Trichopus zeylanicus. Phytotherapy Research, 19, 669–673.

[3] Powers, S., & Lennon, S. L. (1999). Analysis of cellular responses to free radicals: Focus on exercise and skeletal muscle. Proceedings of the Nutrition Society, 58, 1025–1033.

[4] Sehirli, O., Tozan, A., Omurtag, G.Z., Cetinel, S., Contuk, G., Gedik, N., Sener, G. (2008). Protective effect of resveratrol against naphthalene-induced oxidative stress in mice. Ecotoxicol. Environ. Saf., 71, 301-308.

  1. The results are shown with different decimals. Standardize all data with the same decimals along the text (for example Table 1).

Response:Thanks for the reminding. We have unified all the results and data with the same decimals in the Table 1 and the text.

  1. Figure 6: Indicate the unit of measure on the y axis of Figure 6B; detail the legend of the figure explaining in detail what the images 6a, 6b, and 6c represent.

Response:Thanks for the reviewer’s suggestions. We have edited the unit of measure on the y axis of Figure 6B and modified the figure captions so that readers can understand the content of the picture more quickly.

  1. Explain the acronyms of the analyzed samples in the legends of all the images.

Response: Thanks for the nice suggestions. The acronyms of the analyzed samples in the legends of all the images have been added in the revised manuscript. Please check the updated Figure 1, 4, 5, 6 and Table 1.

Minor revisions:

  1. Explain all acronyms the first time you use it in the text, for example: SLNs, SIRT1, LC3-II and p62 in the abstract; “EE” in the image legend of Figure 4 at line 107; “NA” in Table 1 at line 125; “HE” at line 130; MDA” at line 147

Response: Thanks very much for your nice suggestions. All acronyms the first time to use have been added in the revised manuscript. Please check the updated Line 15, 23, 25, 26 in the abstract for SLNs, SIRT1, LC3-II and p62; “EE” in the image legend of Figure 4 at line 113; “NA” changed into “RSV-SLNs“ in Table 1 at line 131; “HE” at line 138; MDA” at line 157. Besides, we added all the missing acronyms in the text, such as, the updated Line 65, 122, 123, 138, 158, 162, 199, and 200.

  1. Missing space: Line 36 after the point “[2-3]. Fatigue”; Line 118 after 0.05 “p < 0.05respectively”; Line 220 after RSV “As”.

Response: According to the reviewer’s suggestions, we have added the missing space in the text as listed above in lines 38, 123 and 239.

  1. Double space: Line 59 “control [16]”. Line 150 “alcohol. As”

Response: Thanks to the reviewers for the carefully checking, we have modified the double space to the single space as listed above in lines 61 and 160.

  1. Should be changed in all text: “in vivo” with “in vivo” in italics; “vs.” with “vs” in italics; “ml” with “mL” lines 287, 288, 292.

Response: Thanks to the reviewers for the carefully checking, we have changed all the “in vivo” with “in vivo” in italics, please check the updated Line 2, 18, 63, 340, and 387; “vs.” with “vs” in italics, please check the updated Line 131, 132, 189; “ml” with “mL”, please check the updated Line 305, 306, 307, 310, 311, and 334.

Reviewer 2 Report

In my opinion manuscript is well prepared, innovative in the subject and fits in the field of Molecules. Only small corrects are needed:

1) please correct abbreviations of figure 1,

2) please correct text in lines 71-83 - there is wrong numbering,

3) figure 5 - I do not see white arrows, please explain or correct.

Author Response

Dear Reviewer,

  Thank you for your valuable and thoughtful comments on our manuscript. According to the comments, we have carefully revised our paper. Some alterations have been made and some more details have been added in the revised form. The corrections of the manuscript are highlighted in red through the text. All pages and lines are numbered consecutively. The corrections and responses to the comments are listed as follows, or please see the attachment:

  1. Please correct abbreviations of figure 1.

Response: Thanks for reviewer’s helpful comments. We have corrected the abbreviations of Figure 1. The corrections of the figure captions are highlighted in red.

  1. Please correct text in lines 71-83 - there is wrong numbering.

Response: Thanks to the reviewers for the carefully checking, we have revised the figure numbering in lines 74-86 of the text.

  1. Figure 5 - I do not see white arrows, please explain or correct.

Response: Thanks for the carefully checking, we have corrected all the labeling symbols of Figure 5. Please check the updated Line 152-154. The details such as, black arrow denotes inflammatory cell infiltrate; triangle refers to myonecrosis; square corresponds to centrally nucleated fibers; and pound sign indicates edema.